RBI-ThPhys-2023-42

# Long-range entanglement and topological excitations

G. Torre,[1] J. Odavić,[1, 2] P. Fromholz,[3] S. M. Giampaolo,[1] and F. Franchini[1]

[1]*Ruđer Bošković Institute, Bijenička cesta 54, 10000 Zagreb, Croatia*

[2]*Dipartimento di Fisica Ettore Pancini, Università degli Studi di Napoli Federico II,*
*Via Cinthia, 80126 Fuorigrotta, Napoli, Italy*

[3]*Department of Physics, University of Basel, Klingelbergstrasse 82, CH-4056 Basel, Switzerland*

(Dated: October 26, 2023)

Topological order comes in different forms, and its classification and detection is an important field of modern research. In this work, we show that the Disconnected Entanglement Entropy, a measure originally introduced to identify topological phases, is also able to unveil the long-range entanglement (LRE) carried by a single, fractionalized excitation. We show this by considering a quantum, delocalized domain wall excitation that can be introduced into a system by inducing topological frustration in an antiferromagnetic spin chain. Furthermore, we study the resilience of LRE against a quantum quench and the introduction of disorder, thus establishing the existence of a phase with topological features despite not being a typical topological order or symmetry-protected one.

## I. INTRODUCTION

At its beginning, many-body quantum physics borrowed directly from classical concepts for the classification of phases [1], thus focusing only on local properties. However, quantum mechanics has an intrinsic non-locality, whose role is increasingly underlined by the development of the theory of entanglement. Such non-locality led today's researchers to come to grips with the notion of topology, long-range correlations, and all the mathematical concepts they entail. It is now clear that the quantum phases of matter are affected not only by the local symmetries (preserved or broken) but also by their topological classes that do not change under smooth deformations and thus capture robust global features.

In gapped systems, topological phases are described in terms of the structure of the energy spectrum and eigenvectors in the continuum limit (such as the Bloch bundle for crystalline materials) [2–4]. Such structures are characterized by using topological invariants, *i.e.* quantities associated with a topological space that takes discrete values, thus not changing under continuous deformations of the space. Over the past ten years, it has been realized that the global structure of a non-trivial topology is related to non-local correlations which are hard to detect using standard correlation functions. Fortunately, quantum information theory has provided appropriate tools to condensed matter physicists, such as the different measures of entanglement that allowed us to reformulate our understanding of the various quantum phases in terms of universal short- and long-range entanglement. More specifically, the topological states display long-range entanglement that is not affected by adiabatic local unitary transformation [5–9].

The entanglement-based paradigm of topological phases, from a microscopical point of view, stems from the robust fractionalization of collective modes into at least two deconfined fractional quasi-particles (spatially distinct, independently propagating, and together carrying an integer charge) which are entangled with one another. When localized on the edges, these collective modes are gapless (like the Majorana zero-modes of the Kitaev wire [10]), and when in the bulk, the now excited modes are gapped (like the anyonic modes of the fractional quantum Hall effect [11]). This reformulation inspired entanglement-based topological criteria to characterize the phases, that are now widely used in numerical simulation [12–24].

Fractionalization is not limited to topological orders but happens commonly in strongly correlated systems and one-dimensional Luttinger liquids in particular [25]. For instance, spinons, *i.e.* excitations carrying spin-1/2 but no electric charges emerge as collective fractionalized excitations in antiferromagnetic chains [26, 27] and so do kink and antikink states in sine-Gordon models [28]. Similarly, a spinful fermion, when injected in a 1D wire, naturally separates into two independently moving excitations, one carrying only charge (holon) and the other only the spin degrees of freedom (spinon) [29]. Unlike in topological phases, these fractional modes are not robust towards generic local perturbations and are more likely to be confined [30], preventing their exclusive properties (fractional charge and fractional statistics) from being easily observed [31–34].

Coming to one-dimensional systems, which is the subject of this work, true, robust topological order is no longer possible. Only the so-called *symmetry-protected topological* (SPT) phases survive. Differently from the true topological order, in SPT phases, topological invariants are unaffected by local deformations as long as these last respect prescribed symmetries. But, in one-dimensional systems, this order is hard to detect. For instance, entanglement entropy is commonly used in higher dimensions because the first finite constant correction beyond the area law, known as *topological entanglement entropy* (TEE), encodes the topological degeneracy of the ground states. In 1D, the area law itself prescribes a leading term not scaling with the subsystem size and thus a topological contribution, if present, gets screened by area one. To overcome this limitation it was proposed

to use the quadri-partite entanglement entropy [2], also called disconnected entanglement entropy (DEE) [35, 36] which is also prospectively relevant for experiments [37–40]. The particular choice of the partitioning in the evaluation of the DEE causes local contributions in the entanglement to cancel out, thus implying that a non-zero value of the DEE can be associated with the presence of long-range correlations typical of topological phases.

In this work, our aim is twofold: on one side, going beyond the limits of both the confinement phenomenon and the context of topological phases, we propose an alternative phenomenon in which, to isolate a fractional state, the energetic protection of topology is replaced by a geometrical frustration at the boundary [41–43]. On the other side, we will show that DEE is sensitive to long-range entanglement beyond the usual SPT phases, such as that of deconfined, fractional topological excitations.

The topological excitation we will utilize for this purpose is the quantum-dressed version of a single delocalized domain wall configuration over a classical Néel state. In a 1D classical ferromagnetic Ising, one can excite a single domain wall excitation by applying a suitable field at the open end of a chain. Such a state is a soliton with a non-zero topological charge (here, given by the difference between the magnetization at the two ends of the chains). In a classical antiferromagnetic Ising, the ground states are typically the two perfectly staggered configurations and their low-energy excitations are domain walls interpolating between the two staggered orders. However, by applying periodic boundary conditions on a chain with an odd number of sites (a setting known as Frustrated Boundary Conditions - FBC) together with antiferromagnetic spin coupling, the Néel states cannot be realized anymore, and the lowest energy configuration is highly degenerate with each possible state hosting a single domain wall. Upon introducing a quantum (non-commuting) interaction (for instance, a transverse field), this huge degeneracy gets lifted, and the ground state typically becomes a superposition of kink states [41]. These kinks are fractionalized excitations and are usually created only in pairs. By increasing the strength of the quantum term, the quantum dressing of the states prevents describing them as containing a single classical kink, but they can still nonetheless be characterized as a delocalized quantum excitation [44]. We will show that, very much like its semi-classical counterpart, this excitation retains its fractionalized nature and thus can be characterized as a topological excitation with long-range entanglement.

Furthermore, we demonstrate the persistence of long-range entanglement after a global quench. In particular, we apply a unitary evolution within both the unfrustrated and frustrated phase of the quantum Ising chain and observe that the disconnected entanglement retains its initial finite or vanishing value (in the limit of an infinite chain) starting from the frustrated and unfrustrated phase, respectively. Such behavior is usually associated with topological invariants in systems that ad-

mit particle-hole symmetry [35, 45]. Finally, we establish the robustness of the observed long-range entanglement by introducing a bond/site disorder into the chain and conclude, in agreement with the results of [42, 46, 47], that perfect translational invariance constitutes a point of transition between the usual Ising phase and a *kink phase* featuring the phenomenology of topological frustration and long-range entanglement [48].

This work is organized as follows. At the beginning of Sec. II we discuss the model that we use throughout our work. This model is the quantum Ising model in the transverse field, which has the advantage of remaining analytically solvable even when FBC are taken into account. Regardless of the analytical solvability of the model, the von Neumann entropy for disconnected subsystems needed for the DEE is hard to evaluate. Therefore, intending to simplify such a problem, in the second part of Sec. II instead of evaluating the Von Neumann entropies of the different contributions, we resort to evaluating their Rényi ones. Then, in Sec. III we move to present our results. First, we derive an analytical expression for the DEE based on the kink state representation of the ground state that is valid close to the classical point (defined as the point in parameter space where all terms in the Hamiltonian commute with one another). Afterward, we show that the numerical evaluation of the DEE in the whole topologically frustrated phase (deep inside the quantum regime) matches the derived analytical result in the thermodynamic limit. Hence, we check the robustness of the observed long-range entanglement, both performing a global quantum quench and studying the system upon introducing localized disorder. Finally, in Sec. IV we conclude from our findings and discuss their significance.

## II. MODEL AND METHODS

As a prototypical quantum magnetic system, we consider the transverse-field Ising chain with $N$ spins described by the Hamiltonian

$$H = J \sum_{k=1}^{N} \sigma_k^x \sigma_{k+1}^x - h \sum_{k=1}^{N} \sigma_k^z, \qquad (1)$$

where $\sigma_k^\alpha$ are Pauli matrices with $\alpha = x, y, z$ on the $k$-th site of the chain, $J$ denotes the spin coupling amplitude and $h$ the magnetic field. Throughout the whole paper, we assume that the system is made of an odd number of lattice sites ($N = 2M + 1$ with $M \in \mathbb{N}$) and satisfy the periodic boundary conditions $\sigma_{N+1}^\alpha \equiv \sigma_1^\alpha$, i.e. that the system holds the so-called FBC. At the classical point ($h = 0$), when ferromagnetic (FM, $J < 0$) interactions are considered, we have a twofold degenerate frustration-free ground state. Conversely, in the presence of anti-ferromagnetic couplings (AFM, $J > 1$), the presence of the FBC forbids the realization of the two Néel states since at least one pair of neighboring spins needs to be

aligned ferromagnetically. As a consequence, the system is frustrated. Hence, geometrical boundary frustration in one dimension at the classical point leads to a $2N$ degenerate ground state manifold, with each state hosting a domain-wall defect in one of the $N$ possible pairs of two neighboring spins aligned in parallel (ferromagnetically), while the factor of two accounts for different Néel orderings.

Entering the quantum regime ($h \neq 0$) this degeneracy is completely lifted and, until $|h|$ becomes greater or equal than 1, the unique ground state becomes part of a band in which the different elements are uniquely identified by their parity and lattice momentum. In this frustrated phase, the energy gap between low-lying states close algebraically as $1/N^2$, and in the thermodynamic limit the frustrated chain becomes gapless, but not relativistic, thus not described by a CFT [46, 49, 50]. This behavior has to be compared with the unfrustrated case, where for $|h| < 1$ we have the magnetically ordered phase with two nearly degenerate lowest energy states (with opposite parity) separated by a gap that closes exponentially with the system size [51, 52], and a finite energy gap with the band of low-energy excited states.

While the properties of the Ising chain without frustration have been investigated for decades, the realization that FBC affects them is quite recent [41–44, 53–55]. In particular, it was realized that in the frustrated phase, some correlations acquire algebraic corrections [46, 49]. For instance, for large $N$, the two-point spin correlation functions along the $x$-direction of the frustrated and the unfrustrated case are related by the following relation

$$\langle \sigma_k^x \sigma_{k+R}^x \rangle^{(f)} \simeq \langle \sigma_k^x \sigma_{k+R}^x \rangle^{(u)} \left(1 - \frac{R}{2N}\right), \qquad (2)$$

where $(f)$ and $(u)$ indices stand respectively for frustrated and unfrustrated.

These corrections in the correlation functions reverberate also in the behavior of the entanglement entropies (EE). In Ref. [44], the von Neumann EE for a subsystem $A$ made of $M$ contiguous sites was calculated for the ground state $|g\rangle$ of the Hamiltonian in (1). The derived expression for von Neumann EE for large $N$ and $|h| < 1$ could be compactly expressed as

$$S_A^{(f)} = S_A^{(u)} - \frac{M}{N} \ln \frac{M}{N} - \left(1 - \frac{M}{N}\right) \ln \left(1 - \frac{M}{N}\right). \quad (3)$$

Here $S_A$ stands for the von Neumann entanglement entropies of the reduced density matrix $\rho_A$

$$\rho_A \equiv -\mathrm{Tr}\left(\rho_A \ln \rho_A\right). \qquad (4)$$

where $\rho_A = \mathrm{Tr}_{A^c} |g\rangle\langle g|$ is obtained by tracing out from $|g\rangle$ all the degrees of freedom in $A^c$, which is the complement of $A$. These results indicate that, compared to the non-frustrated case, in the frustrated one, there is the presence of an extra amount of entanglement that is compatible with the interpretation that its ground state is characterized by an additional, delocalized quasi-particle.

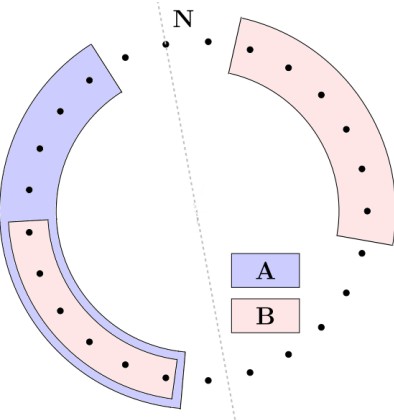

FIG. 1. Sketch of the subsystems used in the disconnected Renyi$-\alpha$ entanglement entropy in Eq. (5). The black dots represent the spins in the chain ($N$ is always chosen to be odd). For what concern the subsystems, one of the two parts of $B$ is always included in $A$, while the other is a subset of $A^c$. Moreover, the subset $A$ and the part of $B$ included in it always share a boundary.

As this excitation's characteristic length scale extends to the entire chain, we aim at extracting the long-range entanglement it entails using the Disconnecting Entanglement Entropy. At first sight, the problem of calculating the disconnected entanglement entropies for the model in eq. (1), of which we have an analytical solution, does not seem particularly challenging. However, the analytical computation of entanglement in disconnected subsets is a rather complex process that has been successfully addressed only in a few cases with very tightening hypotheses [56–61]. On the other side, a numerical evaluation of the von Neumann entropy for a disconnected partition forcibly passes by the determination of the whole set of eigenvalues of the reduced density matrix, which greatly limits the size of the subsets that can be considered. To ease the computational complexity, and to obtain results for larger subsystems, that are crucial for the present work, we employ the disconnected Rényi-$\alpha$ entanglement entropy $S_\alpha^D$ [2] defined as

$$S_\alpha^D = S_{A,\alpha} + S_{B,\alpha} - S_{A\cup B,\alpha} - S_{A\cap B,\alpha}. \qquad (5)$$

The Rényi$-\alpha$ entanglement entropy, for which different numerical techniques are known to efficiently compute it [62–67], for a generic subsystem $A$ with reduced matrix $\rho_A$ is given by

$$S_{A,\alpha} = \frac{1}{1-\alpha} \ln \left[\mathrm{Tr}\left(\rho_A\right)^\alpha\right]. \qquad (6)$$

Here $\alpha \in [0,1) \cup (1,\infty]$ and in the limit $\alpha \to 1$, we recover the von Neumann entanglement entropy in (4). From here on we will focus on the Rényi entropy of order $\alpha = 2$ [68–71] which has always attracted particular attention since it can be observed directly in some experimental set-ups [72–76]. The subsystems $A, B$ considered in the

present work are sketched in Fig. 1, with $B$ always split into two parts of equal size. It is worth noting that, since topological frustration implies the assumption of periodic boundary conditions, the geometry of the two subsets is slightly different with respect to the one considered in Refs. [35, 36] where open boundary conditions were taken into account.

## III. RESULTS

### A. Analytical results close to the classical point

Let us start our analysis by taking into account the ground state close to the classical point, i.e. for $h \to 0$. Assuming ferromagnetic couplings ($J = -1$), the ground state of eq. (1) is represented by a GHZ state, where the two components are the tensor product of the eigenstates of $\sigma_k^x$ with the same eigenvalue. Independently of the subsystems, the reduced density matrices obtained by such a state have two eigenvalues equal to $1/2$, while all the other vanishes and, as a consequence, $S_2^D$ is identically zero.

On the contrary, when $J = 1$, the GS of eq. (1) is well approximated by an equal weight superposition of the $2N$ kink states [54, 77], which are the states obtained by introducing a domain-wall defect in one of the $N$ spin of the chain into one of the two Néel states in the base of the eigenstates of $\sigma_k^x$. Exploiting this expression of the GS, we obtain the reduced density matrix of the system and consequently the analytical expressions for all terms in eq. (5) (see Appendix A for a detailed explanation).

In the case of a subsystem $A$ made of $M = mN$ contiguous spins, in the limit of large $N$ and finite $m$, we find that eq. (3) generalizes to

$$S_{A,2} = -\ln\left[\left(m^2 + (1-m)^2\right)\right] + \ln(2). \qquad (7)$$

On the contrary, in the case in which the subsystem is made of two disconnected parts, one of them made of $M = mN$ spins and the other made of $R = rN$ spins, separated by a distance equal to $L = lN$ (see Fig. (2)) we obtain that when $N$ diverges and $m$, $r$ and $l$ stay finite, the value of the Rényi-$\alpha$ entropy becomes

$$S_{A,2} = -\ln\left[\left(m^2 + l^2 + r^2 + (1-m-l-r)^2\right)\right] + \ln(2) \quad (8)$$

By collecting the different contributions, we obtain the value of $S_2^D$ for the ground state of the frustrated quantum Ising model in the limit $h \to 0$ that we indicate with $\tilde{S}_2^D$. Considering the particular choice depicted in Fig. 1 in which we assume that both the subsystems $A$ and $B$ comprise the same number of sites $M$ and that $B$ is split

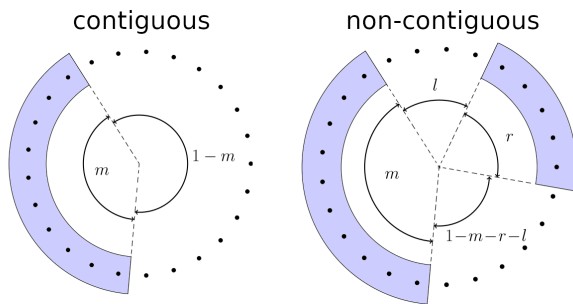

FIG. 2. Generic partitions considered in respect to Eq. (5). We consider a contiguous subset of $m = M/N$ spins in the left panel while, in the right one, the subsystem of interest is divided into disconnected sectors of length $m = M/N$ and $r = R/N$, while being separated by $l = L/N$ spins.

into two parts of equal length, we obtain

$$
\tilde{S}_2^D(m,l) = -\Big[\ln\left(m^2 + (1-m)^2\right) - \qquad (9)
$$
$$
- \ln\left(\left(1 - \frac{m}{2}\right)^2 + \left(\frac{m}{2}\right)^2\right) +
$$
$$
+ \ln\left(l^2 + (1-l-m)^2 + \frac{m^2}{2}\right) +
$$
$$
- \ln\left(l^2 + \left(1 - l - \frac{3m}{2}\right)^2 + \frac{5}{4}m^2\right)\Big],
$$

which, differently from the unfrustrated case, does not vanish. This first simple result immediately marks a difference between a frustrated and an unfrustrated system, and it is associated with the presence of a long-range contribution to the entanglement.

### B. Analytical results far from the classical point

Moving away from the classical point, the ground states of both systems are no longer represented by the simple states that we have analyzed so far. Nevertheless, we can still make some general predictions about the behavior of the DEE, making use of an approach previously exploited in Ref. [78] that derives, essentially, from the Lieb-Robinson bound theorem [79]. In order to proceed, let us recall that Rényi entropy of order 2 of a reduced density matrix $\rho_A$ can be written as $S_{A,2} = -\ln P_A$ where $P_A = \text{Tr}(\rho_A^2)$ is the purity of the reduced density matrix $\rho_A$. To evaluate the purity of $\rho_A$, we can exploit the identity $P_A = \text{Tr}(\rho_A^2) = \text{Tr}(\mathcal{S}_A \rho^{\otimes 2})$ [80] where $\rho^{\otimes 2} = \rho \otimes \rho$ is the tensor product of two copies of $\rho$ and $\mathcal{S}_A$ stands for the swap operator of order two with support on $A \otimes A$. Such an operator can be written as $\mathcal{S}_A = \otimes_{k \in A} \mathcal{S}_k$, where

$$
\mathcal{S}_k \, |i_1, \ldots, i_k, \ldots, i_N\rangle \otimes |j_1, \ldots, j_k, \ldots, j_N\rangle
$$
$$
= |i_1, \ldots, j_k, \ldots, i_N\rangle \otimes |j_1, \ldots, i_k, \ldots, j_N\rangle \quad (10)
$$

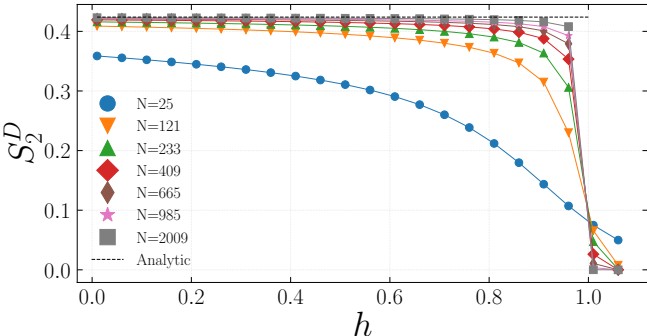

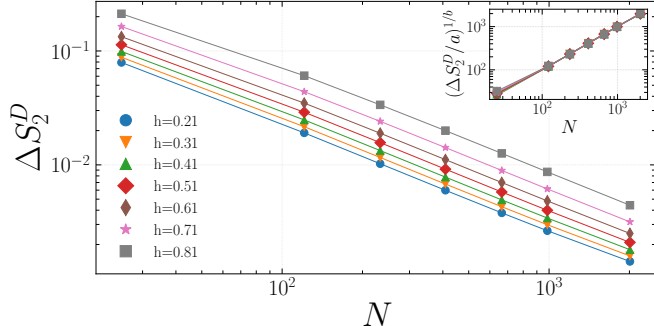

FIG. 3. Disconnected Renyi-2 entanglement entropy as a function of the magnetic field $h$ for different lengths of the system. We fixed the dimension of the subpartition to $m = (N-1)/2N \approx 1/2$ and $l = (N-1)/8N \approx 1/8$. The black horizontal line is the result in Eq. (9), which in this case yields $\tilde{S}_2^D(1/2, 1/8) \simeq 0.423814$ and is the asymptotic value in the thermodynamic limit for the amount of long-range entanglement in the frustrated phase of $0 < h < 1$.

FIG. 4. Behavior of $\Delta S_2^D(N)$ in eq. (12) as a function of the chain length $N$. We observe that in the thermodynamic limit, regardless of the value of $h$, it vanishes with the law $\Delta S_2^D = aN^b$ with $b \simeq -0.935 \pm 0.006$. In the inset, we provide the proof of the generality of the dependence of $\Delta S_2^D(N)$ on $N$, showing the scaling relation $(\Delta S_2^D/a)^{(1/b)} = N$.

Having recalled these concepts, let us turn our attention to the Hamiltonian in (1) that can be seen as the sum of local bounded terms. Assuming $J = -1$, in the magnetically ordered phase for $h < 1$, the spectrum is characterized by a low energy sector composed of two nearly degenerate states identified by their parity and separated from the rest by a finite gap. Therefore, starting from the ground state close to the classical point, we can obtain the ground state in each other point of the phase by the quasi-adiabatic continuation $U(h)$ induced by a continuous deformation of $H(h)$ if such deformation commutes with the parity operator [81]. Due to the deformation, a local operator with support on a subsystem $A$ transforms into an operator with support in the whole system. Nevertheless, the locality of the Hamiltonian implies that we can arbitrarily approximate it with an operator with support in $A'$ that has a diameter equal to $diam(A') = diam(A) + \chi$, as long as $\chi$ is larger than the correlation length of the system $\zeta$. In making this replacement, we made an error bounded from above by the quantity $Ke^{-\chi/\zeta}$ where $K$ is a model-dependent constant.

Let now $\rho(0)$ be the density matrix of the ground state of the unfrustrated system close to the classical point. The purity of the projection of the evolved state $\rho(h)$ to a subset $A$ reads

$$
\begin{aligned}
P_A(h) &= \mathrm{Tr}\left[\rho(h)^{\otimes 2}\mathcal{S}_A\right] \\
&= \mathrm{Tr}\left[U(h)^{\otimes 2}\rho(0)^{\otimes 2}(U^\dagger(h))^{\otimes 2}\mathcal{S}_A\right] \\
&= \mathrm{Tr}\left[\rho(0)^{\otimes 2}(U^\dagger(h))^{\otimes 2}\mathcal{S}_A U(h)^{\otimes 2}\right] \quad (11) \\
&\approx \mathrm{Tr}\left[\rho(0)^{\otimes 2}\mathcal{S}_{A+\chi}\right],
\end{aligned}
$$

where $\mathcal{S}_{A+\chi}$ denotes the swap operator with support on spins that are, at most, at distance $\chi$ from $A$. Therefore, the purity of the reduced density matrix obtained from $\rho(h)$ by tracing out all degrees of freedom outside $A$, is

well approximated by the purity of that obtained from $\rho(0)$ by tracing out the complement of $A + \chi$ until $\chi$ is much greater than the correlation length $\zeta$ of the system. But, as we have seen previously, this second purity is equal to $1/2$ regardless of the choice of $A$ and hence the fact that the DEE of the unfrustrated system vanishes even for $h \neq 0$.

Turning back to the frustrated case, instead of having a ground state manifold composed of only two states, we have a band composed of $2N$ states, each of which is uniquely identified by its momentum and parity. Therefore, if we consider a deformation of the Hamiltonian that is translational invariant and commutes with the parity along the Z axis, we can still obtain the density matrix of the system at $h \neq 0$ from the quasi-adiabatic continuation of the density matrix $\rho(0)$ of the ground state close to the classic point. Hence, eq. (11) remains valid also in the case of frustrated systems. However, in this case, as we have already seen, the value of the purity of $\rho_A$ depends on its size and whether it is, or is not, made of contiguous spins. But, in the limit in which all the parameters of the subsystems, i.e. $M$, $R$, and $L$, grow proportionally to $N$, the eigenvalues converge to a thermodynamic limit (see Appendix A). Therefore, in this limit, the DEE for the frustrated ground state becomes independent of $h$ until the phase transition is reached.

## C. Numerical results for the finite spin chain

Exploiting the fact that the transverse-field Ising chain in (1) remains integrable even in the presence of frustration [41, 49, 52, 82, 83], we can directly test the results obtained in the previous sections. For the sake of simplicity, from now on, we will focus on the particular case in which both $A$ and $B$ are made of $M = mN = (N-1)/2$ spins and $L = lN = (N-1)/8$.

The results of our tests are summarized in Fig. 3 where

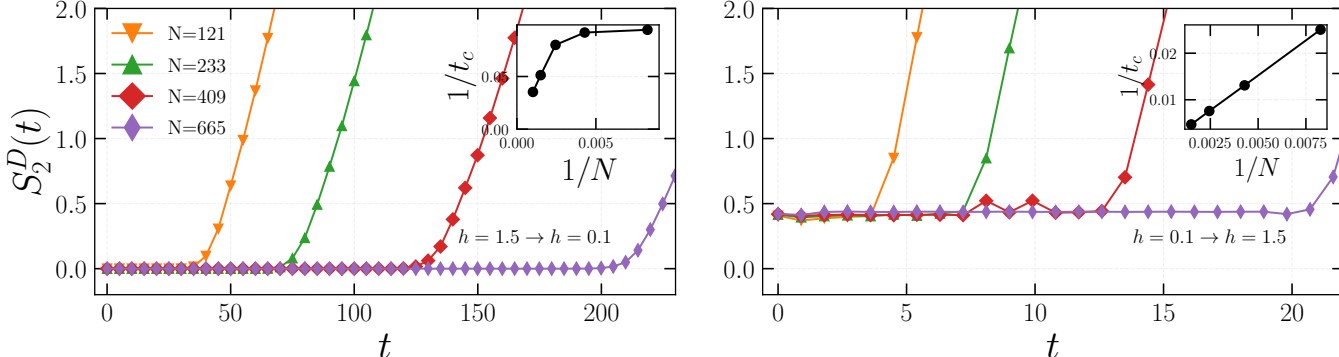

FIG. 5. Time evolution of the disconnected Rényi-2 entropy after a global quantum quench. *Left panel* $S_2^D$ after quenching the Hamiltonian from the paramagnetic phase ($h = 1.5$) to the frustrated one $h = 0.1$ as a function of time and for different system sizes $N$. *Right panel* $S_2^D$ after quenching the Hamiltonian from the frustrated phase $h = 0.1$ to the unfrustrated one $h = 1.5$ as a function of time and for different system sizes $N$. In both cases, $S_2^D$ remains constant up to the critical time $t_c$ after the quench that depends on the size of the system and diverges with $N$, as it can be seen by the insets. In all the simulations, we have used $m = (N-1)/2N \approx 1/2$, and $l = (N-1)/8N \approx 1/8$.

we plot the values for the disconnected Rényi-2 entanglement entropy as a function of the magnetic field $h$, for different sizes $N$ of the chain. Confirming our expectations, in the frustrated phase $|h| < 1$, it differs from zero and tends to $\tilde{S}_2^D(m, l)$ in eq. (9) going towards the thermodynamic limit, with a slower convergence the closer one gets to the phase transition where the correlation length of the system diverges. These behaviors should be confronted with what we have in the absence of frustration, where the area law imposes the vanishing of the DEE [62]. The insensitivity of the disconnected Rényi-2 entanglement entropy for the microscopic details of the model is a strong indicator of some sort of topological origin. However, in our case, the value of $\tilde{S}_2^D$ is not always quantized to the same integer number as in Ref. [36], but its value is fixed by the geometry of the partition. This is due to the fact that, in this case, the long-range entanglement does not just correlate the subsystems' boundaries but extends through the entire chain.

In Fig. 4 we plot the differences between the value of the disconnected Rényi-2 entanglement entropy and its thermodynamic limit $\tilde{S}_2^D$ in the frustrated phase as a function of $N$, for different $h$, i.e. the quantity

$$\Delta S_2^D(N) = \tilde{S}_2^D(1/2, 1/8) - S_2^D(N, (N-1)/2, (N-1)/8). \quad (12)$$

We can see that the Rényi-2 disconnected entanglement entropy approaches its thermodynamic value with a power-law $aN^b$ where $b \simeq -0.935 \pm 0.006$ is independent of microscopic details of the model, such as the value of the external field. This behavior is consistent with the presence of non-local extensive entanglement properties of the frustrated systems.

*Effects of a global quench*

The fact that throughout the frustrated phase, the value of the disconnected Rényi-2 entanglement entropy does not depend on $h$ supports the idea that it is induced by topological effects. To provide further support to this hypothesis, let us analyze the response of the DEE to a global quench of the Hamiltonian parameters of the system. It is known that, in the thermodynamic limit, quantities associated with both topological orders and particle-hole-protected topological phases are not sensitive to this type of change [84, 85]. Moving from the thermodynamic limit to the case of systems of finite size, these quantities begin to respond to a sudden quench of the Hamiltonian parameters only after a certain delay time, which increases as $N$ increases.

Hence, let us consider the evolution of $S_2^D$ after a global quantum quench protocol [86–88]. We consider the system being in the ground state of (1) with initial magnetic field $h_i$ and, at $t = 0$, in the whole chain we suddenly change the magnetic field to $h_f$. The quench drives the system far from the equilibrium and hence its state starts to evolve driven by the new Hamiltonian. In particular, we force the system across the phase transition, that is, we evolve the ground state of the frustrated phase with a Hamiltonian corresponding to a non-frustrated phase and vice versa.

The results are summarized in Fig. 5. It is easy to appreciate that, regardless of its initial value, the disconnected Rényi-2 entropy for finite systems does not change until a size-dependent critical time $t_c$ is reached. To make our analysis more quantitative, following the approach of Ref. [35], we define $t_c$ as the first time value for which the relative change of $S_2^D$ reaches 10%:

$$t_c := \min_t \left\{ t \in [0, \infty) : \frac{|S_2^D(0) - S_2^D(t)|}{S_2^D(0)} > 0.1 \right\}. \quad (13)$$

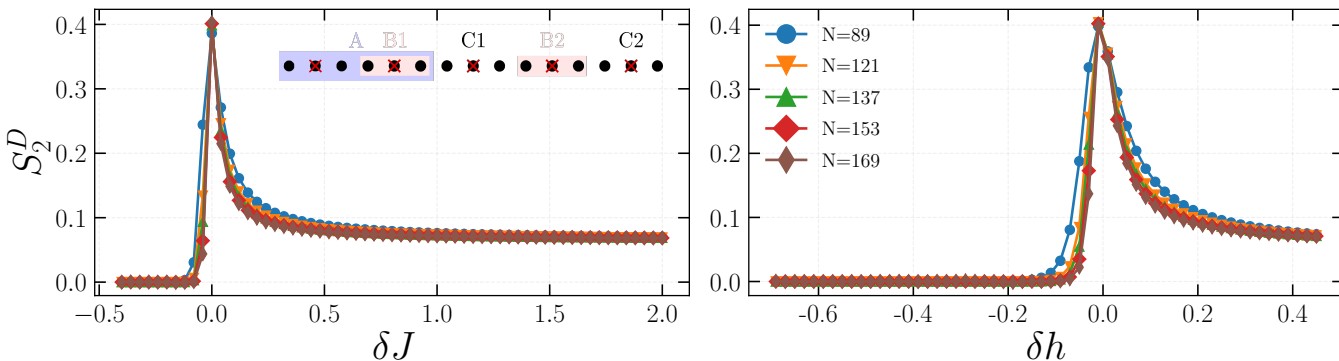

FIG. 6. *Left*: Renyi-2 disconnected entanglement entropy $S_2^D$ as a function of the bond disorder strength $\delta J$ for $\delta h = 0$ *Right*: Renyi-2 disconnected entanglement entropy $S_2^D$ as a function of the field disorder strength $\delta h$ for $\delta J = 0$ In both cases, we have assumed $J = 1$ and $h = 0.5$ while we have placed the defect in the center of the $A/B1$ subsystem.

In the insets of Fig. 5, we plot $1/t_c$ as a function of $1/N$ for the two cases. We observe that when $N \to \infty$, the characteristic time always diverges.

This behavior is the same as that of quantities regarded as topological invariants of topological orders and particle-hole-protected topological phases. More specifically, the behavior of the disconnected Rényi-2 entanglement entropies that we have obtained matches perfectly the one shown in Refs. [35, 45].

### *Effects of disorder*

In Ref. [43] it has been demonstrated that the phenomenology of the frustrated phase is robust against the presence of defects of the antiferromagnetic type, while is wiped off by the presence of ferromagnetic ones. Therefore, it is interesting to analyze if and how the DEE can also detect this dependence on the sign of the strength of the defect. Thus, we generalize Eq. (1) introducing two

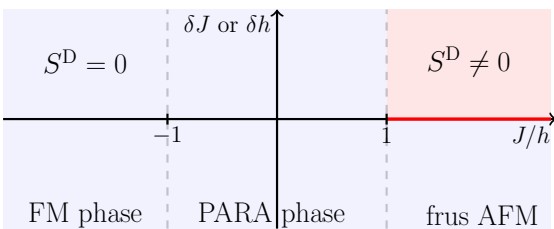

FIG. 7. Phase diagram of the geometrically frustrated Ising spin chain in Eq. (14) with respect to a single bond or site disorder $\delta J$ and $\delta h$, respectively. The disconnected entanglement entropy $S^D$ shows a non-vanishing value (red color) in the thermodynamic limit (TD) for positive disorder strengths. Additionally, the red line denotes where the exact results are obtained. See Sec. II and Eq. 14 for more details about considered model.

different types of localized defects as

$$H = J \sum_{k=1}^{N} \sigma_k^x \sigma_{k+1}^x - h \sum_{k=1}^{N} \sigma_k^z + \delta h \sigma_q^z + \delta J \sigma_q^x \sigma_{q+1}^x. \quad (14)$$

In eq. (14) we place a defect on the $q$-th bond and on the $q$-th site, but we will consider only cases in which at least one of the defect strength is set to zero. A more general case is treated in the Appendix B, showing that the picture we are about to describe is robust. Even in the presence of one or more defects, the Hamiltonian can be solved by resorting to the same techniques used in the ideal case [43, 52, 82]). The results that we have obtained are summarized in Fig. 6.

Let us first set the site defect to zero ($\delta h = 0$) and consider the effect of a bond defect on the anti-ferromagnetic chain ($J = 1$). As shown in the left panel of Fig. 6, disorder strength $\delta J < 0$ tends to favor a ferromagnetic alignment on that bond and localize the excitation, hence destroying the frustrated phase. Accordingly, we observe that DEE vanishes. An AFM defect $\delta J > 0$ instead keeps the DEE finite, and we observe that it tends to a finite, constant value in the limit of a strong defect. These observations are in agreement with one made in Refs. [43, 46] where it was argued that perfect translational invariance without defect constitutes a quantum phase transition separating the Ising phase with no long-range entanglement for $\delta J < 0$ from a kink phase characterized for $\delta J > 0$. While in Ref. [88] the two phases were distinguished by their gap properties, and in Ref. [43] by a different behavior of the order parameter, here we provide a different characterization of these two phases in terms of long-range entanglement or lack thereof.

Similar results are also obtained by switching off the bond defect $\delta J = 0$ and turning on a finite magnetic defect $\delta h$. These results can be appreciated in the right panel of Fig. 6 where we observe that when the defect in the magnetic field favors the localization of the excitation on that particular site the DEE vanishes, while it remains finite for $\delta h > 0$. The two plots in the figure

were obtained by placing $q$ in the middle of the $A/B1$ set of spins. In general, one may expect that the behavior of the DEE would be dependent on the relative position of the defect with respect to the subsets $A$ and $B$. However, all our results show similar behaviors.

The physical picture emerging from all these results about the DEE of the ground state of the Hamiltonian in (14) is summarized in the phase diagram in Fig. 7. Additionally, in Appendix B, we perform the same calculations in the case of two bond disorders and observe a consistent phenomenology, which further supports the existence of an extended phase of frustration and long-range entanglement.

## IV.  DISCUSSION & CONCLUSIONS

We have shown that the disconnected entanglement entropy (DEE), originally introduced to detect (symmetry-protected) topological phases, is also able to reveal the presence of the long-range entanglement generated by a single, delocalized topological excitation. To inject such excitation into a system, we considered an antiferromagnetic chain with an odd number of spins and periodic boundary conditions, also known as frustrated boundary conditions (FBC). Close to the classical point (that is the point, where the strength of the external field vanishes and the Hamiltonian reduces to a sum of commuting terms), the ground state of such systems can be written down as an equal-weight superposition of kink states (domain wall configurations). This simplified picture of the kink states allowed us to derive an analytical expression of the DEE that is purely geometric and reflects how the system is partitioned into subsystems. Thus, in the case we considered, DEE is not always quantized to the same value. This is because the long-range entanglement caused by the topological excitation is spread over the entire chain and not just localized, for instance, at the boundaries. Employing the framework of the quantum Ising model, we provide numerical proof that, in the thermodynamic limit, the analytically derived formula remains valid and correctly captures the long-range entanglement in the DEE in the entire frustrated phase. Furthermore, we checked the robustness of the observed long-range entanglement against the addition of AFM defects in the chain and the action of a global quantum quench.

It is worth noting that the one-dimensional quantum Ising model with antiferromagnetic interaction and imposed FBC does not display a typical symmetry-protected topological phase. In fact, it lacks a gap and/or an apparent protecting symmetry. On the other hand, it cannot be included in the *gapless* topological phases in-

troduced either in Ref. [89], since no conformal field theory may describe the frustrated phase whose low-energy excitations have a Galilean spectrum, or in Ref. [90], since frustration does not induce protected edge modes. Thus, our results support the idea that the geometrically frustrated boundary systems represent a new and different family of topological models characterized by a non-relativistic gapless spectrum and deconfined fractionalized excitation. In traditional STP phases it is also possible to introduce indices or invariants, which capture the topological properties of these systems, while we cannot yet provide such characterization for systems with long-range entanglement due to topological excitations. Nonetheless, based on the considerations above, we think that it is justified to call these systems *"topologically frustrated"* as it has been done so far in the literature [41, 53–55].

From another point of view, the finiteness of DEE in topologically frustrated systems indicates that it can detect long-range entanglement beyond the (symmetry-protected) topological order. In this respect, it would be interesting to calculate DEE in other systems supporting fractionalized/topological excitations and check whether the quantization formula we derived here remains valid or has to be modified. Moreover, providing a path to directly characterize the long-range nature of the correlations in terms of topological invariants or indices is an intriguing avenue for future research. By doing so, we can deepen our understanding of the fundamental principles governing the behavior of topological excitations in condensed matter systems and pave the way for the development of novel materials and technologies with exotic properties.

## ACKNOWLEDGMENTS

We thank Marcello Dalmonte for the numerous discussions which helped us in the development of this work. S.M.G., F.F., and G.T. acknowledge support from the QuantiXLie Center of Excellence, a project co–financed by the Croatian Government and European Union through the European Regional Development Fund – the Competitiveness and Cohesion (Grant KK.01.1.1.01.0004). F.F. and S.M.G. also acknowledge support from the Croatian Science Foundation (HrZZ) Projects No. IP–2019–4–3321. J.O. recognizes both the support of the Croatian Science Foundation under grant number HRZZ-UIP-2020-02-4559 and of the 376 PNRR MUR Project No. PE0000023-NQSTI. P.F. and part of this work were supported by the Swiss National Science Foundation, and NCCR QSIT (Grant number 51NF40-185902).

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

**Appendix A: Rényi-$\alpha$ entropies for the GS of the topologically frustrated system near the classical point**

Near the classical point $(h \to 0^+)$ the GS $|g\rangle$ of the Hamiltonian in eq. (1) is well described by an equal weight superposition of $2N$ *"kink"* states.

$$|g\rangle = \frac{1}{\sqrt{2N}} \sum_{k=1}^{N} \left( |k^+\rangle + |k^-\rangle \right), \tag{A1}$$

where the family of states $|k^+\rangle$ and $|k^-\rangle$ are defined as

$$|k^+\rangle = T^{k-1} \bigotimes_{j=1}^{N'} \sigma_{2j}^z |+\rangle^{\otimes N}$$

$$|k^-\rangle = T^{k-1} \bigotimes_{j=1}^{N'} \sigma_{2j}^z |-\rangle^{\otimes N}. \tag{A2}$$

In this definition of the kink states $|\pm\rangle$ denotes a positive/negative spin in the $x$-direction, $N' = (N-1)/2$, while $T$ is the translation operator that shifts the state of the system by one site towards the right. For $k = 1$ the ferromagnetic defect is placed between site 1 and the spin on site $N$ while with $k > 1$ the translation operator moves it around the whole chain.

From the expression of the ground state in eq. (A1) we may compute the entanglement entropy in the cases of a single interval and of two disconnected regions separately.

Let us start with the simplest case, where the subsystem is made by $M$ contiguous spins. In this case, given $M$ the number of spins making of the partition $A$, the reduced density matrix can be written as follows.

$$\rho_A = \frac{1}{N} \begin{pmatrix} \mathbf{1}_{a,a} & \mathbf{0}_{a,a} & \mathbf{1}_{a,1} & \mathbf{1}_{a,1} \\ \mathbf{0}_{a,a} & \mathbf{1}_{a,a} & \mathbf{1}_{a,1} & \mathbf{1}_{a,1} \\ \mathbf{1}_{1,a} & \mathbf{1}_{1,a} & \frac{N-a}{2} & 2 \\ \mathbf{1}_{1,a} & \mathbf{1}_{1,a} & 2 & \frac{N-a}{2} \end{pmatrix} \tag{A3}$$

In eq. (A3) $a = M - 1$ and $\mathbf{1}_{l,m}$ ($\mathbf{0}_{l,m}$) stands for a matrix with $l$ rows and $m$ columns in which all the elements are equal to 1 (0). This expression is obtained on a basis made by $2M$ states, of which: 1) the first $a$ states are obtained by tracing out the degree of freedom of $A^c$ from the states $|k^+\rangle$ when $k \in A$ and it is an odd number and from the states $|k^-\rangle$ when $k \in A$ and it is even; 2) Similarly, the next $a$ states are obtained from the states $|k^+\rangle$ when $k \in A$ and it is an even number and from the states $|k^-\rangle$ when $k \in A$ and it is odd; 3) the last two states are the two different Néel states defined on $A$.

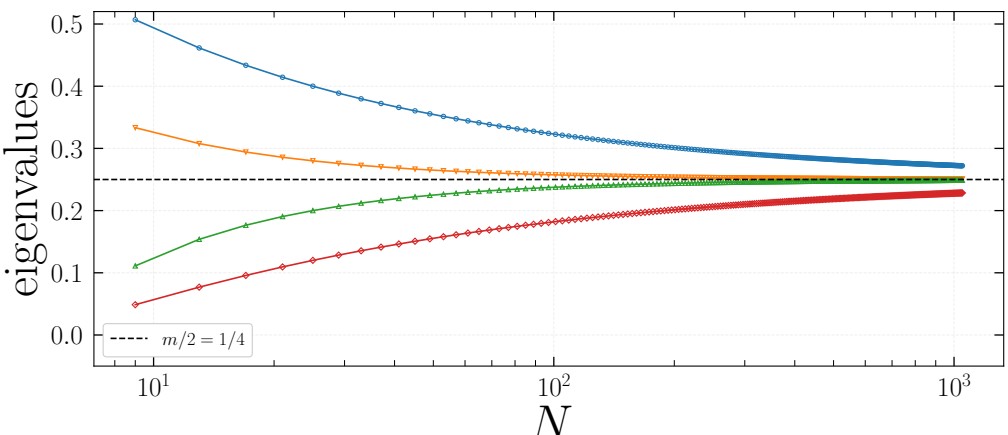

FIG. 8. Non-vanishing eigenvalues of the reduced density matrix in eq. (A3) as a function of the size of the chain for $M = mN = (N-1)/2$. In this case, in the thermodynamic limit, all the eigenvalues tend to 1/4.

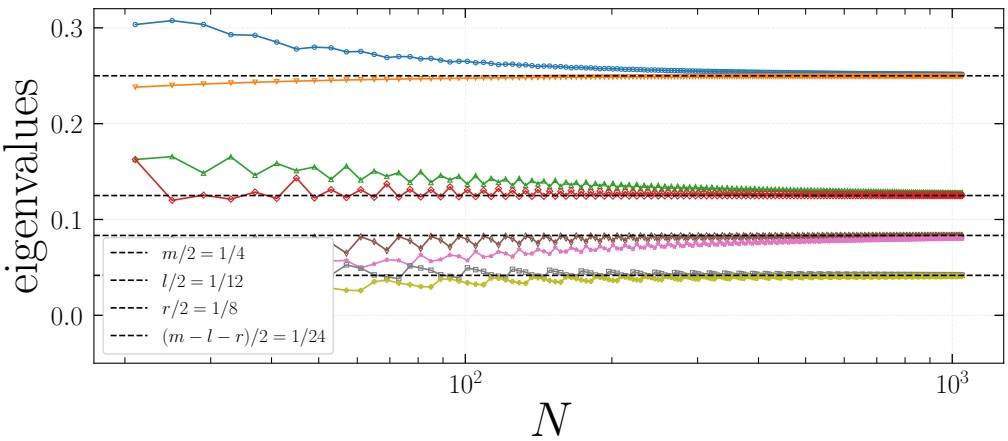

FIG. 9. Non-vanishing eigenvalues of the reduced density matrix in eq. (A5) as a function of the size of the chain for $M = mN = (N-1)/2$, $R = rN = (N-1)/4$, and $L = lN = (N-1)/6$. In this case, in the thermodynamic limit, two eigenvalues admit as limit $1/4$ and two of the other six tend to $1/8$. Of the last four, two tend towards $1/12$, and the last two admit as their limit $1/24$.

Diagonalizing the matrix in (A3) we can see that, except for four of them, all the eigenvalues vanish exactly. In Fig. 8 we depict the behavior of the four non-zero eigenvalues as a function of $N$ in the case in which $M = (N-1)/2$. In general, in the limit of large $N$ and large $M$ we have that two of the four eigenvalues tend to $M/2N$ while the others go toward $(N-M)/2N$. Therefore, denoting $m = M/N$, we recover that the Rényi entropy of order $\alpha$ for large $N$ becomes

$$S_{A,\alpha} = \frac{1}{1-\alpha} \ln \left[ 2^{1-\alpha} \left( m^\alpha + (1-m)^\alpha \right) \right] \tag{A4}$$

Going to the case of a subsystem $A$ split into two parts of which one is made of $M = a+1$ spins, the other of $R = b+1$ spins, and a distance between the two parts equal to $L$, we have that $\rho_A$ can be written as

$$\rho_A = \frac{1}{2N} \begin{pmatrix} \mathbf{1}_{a,a} & \mathbf{0}_{a,a} & \mathbf{0}_{a,b} & \mathbf{0}_{a,b} & \mathbf{1}_{a,1} & \mathbf{0}_{a,1} & \mathbf{1}_{a,1} & \mathbf{0}_{a,1} \\ \mathbf{0}_{a,a} & \mathbf{1}_{a,a} & \mathbf{0}_{a,b} & \mathbf{0}_{a,b} & \mathbf{0}_{a,1} & \mathbf{1}_{a,1} & \mathbf{0}_{a,1} & \mathbf{1}_{a,1} \\ \mathbf{0}_{a,b} & \mathbf{0}_{a,b} & \mathbf{1}_{b,b} & \mathbf{0}_{b,b} & \mathbf{0}_{b,1} & \mathbf{0}_{b,1} & \mathbf{1}_{b,1} & \mathbf{1}_{B,1} \\ \mathbf{0}_{a,b} & \mathbf{0}_{a,b} & \mathbf{0}_{b,b} & \mathbf{1}_{b,b} & \mathbf{1}_{b,1} & \mathbf{1}_{b,1} & \mathbf{0}_{b,1} & \mathbf{0}_{b,1} \\ \mathbf{1}_{1,a} & \mathbf{0}_{1,a} & \mathbf{0}_{1,b} & \mathbf{1}_{1,b} & L+1 & 0 & 1 & 1 \\ \mathbf{0}_{1,a} & \mathbf{1}_{1,a} & \mathbf{0}_{1,b} & \mathbf{1}_{1,b} & 0 & L+1 & 1 & 1 \\ \mathbf{1}_{1,a} & \mathbf{0}_{1,a} & \mathbf{1}_{1,b} & \mathbf{0}_{1,b} & 1 & 1 & c & 0 \\ \mathbf{0}_{1,a} & \mathbf{1}_{1,a} & \mathbf{0}_{1,b} & \mathbf{1}_{1,b} & 1 & 1 & 0 & c \end{pmatrix} \tag{A5}$$

where $c = N + 1 - M - R - L$.

Diagonalizing the matrix in (A5) we can see that, except for eight of them, all the eigenvalues are zero. In Fig. 9 we depict the behavior of the eight non-zero eigenvalues as a function of $N$ in the case $M = R = (N-1)/4$ and $L = N - 1/8$. In general, in the limit of large $N$, $M$, $R$, and $L$ we have that two of the four eigenvalues tend to $M/2N = m/2$, two to $R/2N = r/2$, two to $L/2N = l/2$ while the others go toward $(N - M - R - L)/2N$. Therefore, we compute that the Rényi entropy of order $\alpha$ becomes

$$S_{A,\alpha} = \frac{1}{1-\alpha} \ln \left[ 2^{1-\alpha} \left( m^\alpha + r^\alpha + l^\alpha + (1-m-r-l)^\alpha \right) \right] \tag{A6}$$

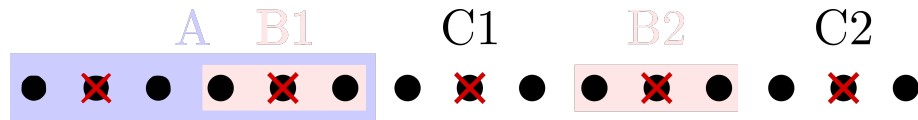

FIG. 10. Partitioning used in the calculation of the disconnected entanglement entropy. Red crosses denote the location of the defects.

## Appendix B: The case of two bond defects

Let us now examine the effects of the presence of two bond defects on the disconnected entropy for the frustrated quantum Ising chain. The Hamiltonian that describe such setting is

$$H = J \sum_{k=1}^{N} \sigma_k^x \sigma_{k+1}^x - h \sum_{k=1}^{N} \sigma_k^z + \delta J_1 \sigma_q^x \sigma_{q+1}^x + \delta J_2 \sigma_p^x \sigma_{p+1}^x. \tag{B1}$$

We consider the various cases in which the two defects are placed in different sub-partitions as illustrated in Fig. 10. The results, obtained by exploiting the methods described in Refs. [43, 52, 82] are summarized in Figs. 11 12, 13, and 14 as a function of the two bond defects strength. Consistently with what we observed in section III C and with the general results in [43, 46], antiferromagnetic defects preserve a finite DEE. However, somewhat surprisingly, there are cases in which the DEE survives even in the presence of one or two ferromagnetic defects.

In Figs. 11 12, 13 the two defects belong to different subsystems, and we observe that the DEE always remains finite when both defects increase the anti-ferromagnetic interaction ($\delta J_{1,2} > 0$), in agreement with the observations regarding the single defect in the main text and in particular Fig. 6. In addition, we observe a finite DEE when both defects are ferromagnetic with the exact same strength. In fact, in this case, instead of localizing the excitation around one of the two equivalent defects, the system creates a Bell pair state shared between them. In such a state, which is equivalent to an edge state, the long-range entanglement of the Hamiltonian without defects, localizes between the two impurities and this explains the observed signal.

Furthermore, in Fig. 14 we consider the cases in which both defects are placed in the same partition (at neighboring bonds) and notice that the DEE is non-zero even in the presence of one ferromagnetic defect, provided that the neighboring defect is anti-ferromagnetic and sufficiently strong, as shown by the extension of the colored area in the $\delta J_1 < 0$ and $\delta J_2 < 0$ region, where it used to vanish in Figs. 11, 12, and 13. This happens because the two neighboring defect contribute in opposite ways toward the attraction/repulsion of the excitation, which, having a characteristic size of the correlation length, feel the contribution from both of them and thus remains delocalized.

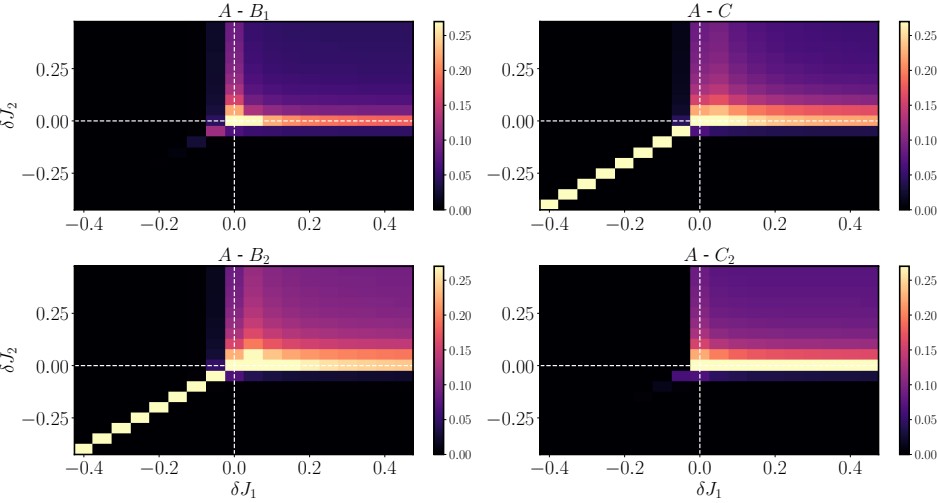

FIG. 11. Density plot of the disconnected Renyi-2 entropy computed for the AFM Ising chain ($J = 1$) with two bond defects and transverse magnetic field $h = 0.5$ for a chain with $N = 121$ spins. The $\delta J_1$ defect is in the $A$ subpartition. White dashed lines guide the eye and indicate where each defect turns from favoring a ferromagnetic alignment on that bond to an AFM one.

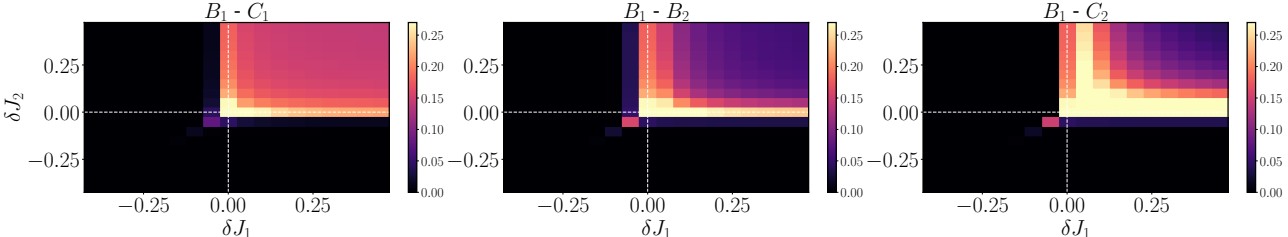

FIG. 12. Density plot of the disconnected Renyi-2 entropy computed for the AFM Ising chain ($J = 1$) with two bond defects and transverse magnetic field $h = 0.5$ for a chain with $N = 121$ spins. The $\delta J_1$ defect is in the $B_1$ subpartition. White dashed lines guide the eye and indicate where each defect turns from favoring a ferromagnetic alignment on that bond to an AFM one. We observe a non-zero DEE when both defects favor an antiferromagnetic alignment ($\delta J_1, \delta J_2 > 0$).

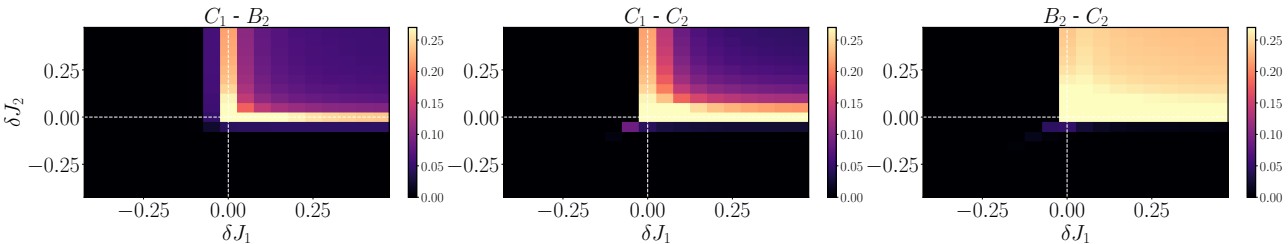

FIG. 13. Density plot of the disconnected Renyi-2 entropy computed for the AFM Ising chain ($J = 1$) with two bond defects and transverse magnetic field $h = 0.5$ for a chain made of $N = 121$ spins. The $\delta J_1$ defect is in the $C_1$ subpartition (first two panels from the left) and in the $B2$ ones (last panel from the left). White dashed lines guide the eye and indicate where each defect turns from favoring a ferromagnetic alignment on that bond to an AFM one. We observe a non-zero DEE when both defects favor an antiferromagnetic alignment ($\delta J_1, \delta J_2 > 0$).

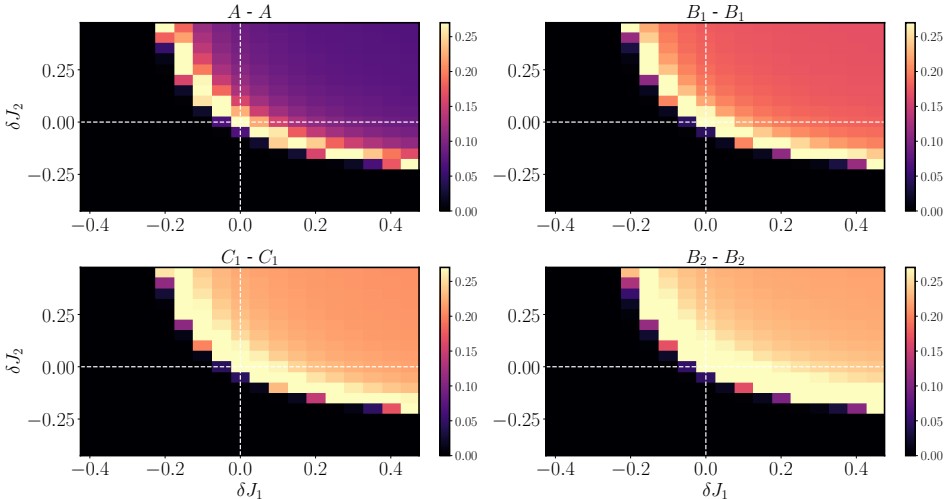

FIG. 14. Density plot of the disconnected Renyi-2 entropy computed for the AFM Ising chain ($J = 1$) with two bond defects placed in the middle of the same sub-partition at neighbor sites, and transverse magnetic field $h = 0.5$. White dashed lines guide the eye and indicate where each defect turns from favoring a ferromagnetic alignment on that bond to an AFM one. The DEE is non-zero even in the presence of one ferromagnetic defect, provided that the neighboring one is anti-ferromagnetic and enough strong.