# Peer review of "Long-range entanglement and topological excitations"

_SciPost Physics_

## Round 1 · Referee Report · Anonymous (Referee 1) · 2024-1-3

Strengths

1 - well written
2 - contains analytical calculations supported by numerics

Weaknesses

It is not clear what is the essential new physics explored in the manuscript.

Report

The present manuscript is focused on the application of the disconnected entanglement entropy (DEE) for the identification of the topological phase in the quantum Ising chain with the frustrated boundary conditions (FBC). The authors demonstrated that the disconnected Rényi-2 entropy can be used as a topological invariant in the model. They obtained an exact expression for this quantity in the classical limit, i.e. in a zero field h=0. This result is supported by the numerical calculations. In addition, the time evolution of the disconnected Rényi-2 entropy after a global quench has been calculated. The effect of defects on a single bond or site is also studied. The authors explained how it may influence the considered entropy and the topological phase.

As I understand, the authors suggested an alternative way to identify the topological phase in a model with the frustrated boundary condition using the DEE. The fact that the DEE behaves like a topological invariant was shown in Ref.36 for the Su-Schrieffer-Heeger model. Here the different model is considered with the specific frustrated boundary conditions. But I am not sure that it reveals the essential new physics. Therefore, I think that the manuscript is more suitable for the SciPost Physics Core.

I have several questions regarding the procedure and the presented results, which would be nice to clarify:

1) In Fig.5 S_2^D(t) goes to much higher values after a global quench than in the equilibrium state in Fig. 3. Could the authors comment on this discrepancy?

2) Eq.(13) contains S_2^D(0) in the denominator. But S_2^D(0) in the left panel in Fig.5 is zero. It seems to me that Eq.(13) should be somehow corrected.

3) S_2^D in Eq.(9) is valid in the classical point h=0. However, since it is a topological invariant it should be constant in the topological phase. Can it be proved at least for small fields?

4) In Fig. 6 the authors show S_2^D as a function of the bond disorder strength $\delta J$, and the field disorder strength $\delta h$. For large values of the disorder parameters S_2^D seems to tend to the same asymptotic value. I am curious if it is possible to find this value analytically, analogously as it was done in Eq. (9).

Requested changes

1) a misprint in Eq.(4): rho_A --> S_A

2) a misprint on page 8: STP phases --> SPT phases

3) In the last paragraph of Sec.IV, the authors call Eq.(9) as a "quantization formula". I am not sure it is completely correct, but I would suggest to add the reference to Eq.(9) there anyway.

---

## Editorial Decision

resubmitted